# The Evolution and Application of a Novel DNA Aptamer Targeting Bone Morphogenetic Protein 2 for Bone Regeneration

**DOI:** 10.3390/molecules29061243

**Published:** 2024-03-11

**Authors:** Mengping Liu, Andrew B. Kinghorn, Lin Wang, Soubhagya K. Bhuyan, Simon Chi-Chin Shiu, Julian A. Tanner

**Affiliations:** 1School of Biomedical Sciences, Li Ka Shing Faculty of Medicine, The University of Hong Kong, Pokfulam, Hong Kong SAR, China; liump3@163.com (M.L.); kinghorn@hku.hk (A.B.K.); wanglin9@connect.hku.hk (L.W.); soubhagyabhuyan@gmail.com (S.K.B.); simon.shiu@hku.hk (S.C.-C.S.); 2Advanced Biomedical Instrumentation Centre, Hong Kong Science Park, Shatin, New Territories, Hong Kong SAR, China

**Keywords:** BMP-2, DNA aptamer, collagen scaffold, regenerative medicine, bone regeneration

## Abstract

Recombinant human bone morphogenetic protein 2 (rhBMP-2) is an FDA-approved growth factor for bone regeneration and repair in medical practice. The therapeutic effects of rhBMP-2 may be enhanced through specific binding to extracellular matrix (ECM)-like scaffolds. Here, we report the selection of a novel rhBMP-2-specific DNA aptamer, functionalization of the aptamer in an ECM-like scaffold, and its application in a cellular context. A DNA aptamer BA1 was evolved and shown to have high affinity and specificity to rhBMP-2. A molecular docking model demonstrated that BA1 was probably bound to rhBMP-2 at its heparin-binding domain, as verified with experimental competitive binding assays. The BA1 aptamer was used to functionalize a type I collagen scaffold, and fraction ratios were optimized to mimic the natural ECM. Studies in the myoblast cell model C2C12 showed that the aptamer-enhanced scaffold could specifically augment the osteo-inductive function of rhBMP-2 in vitro. This aptamer-functionalized scaffold may have value in enhancing rhBMP-2-mediated bone regeneration.

## 1. Introduction

Bone damage and bone defects lead to poor life quality due to a variety of reasons, including pain, continuous infections, swelling at injured sites, and excessive bleeding [1,2]. Unattended severe bone impairments can progress to a disability, diabetes mellitus, and even mortality [3]. In clinical practice, bone grafting is the prevalent strategy used to treat bone fractures. However, grafting is frequently hampered by limited graft resources and donor-site morbidity [4]. Tissue engineering strategies are being developed as promising alternatives to bone grafting, often integrating growth factors such as bone morphogenetic protein 2 (BMP-2) [5,6].

Recombinant human BMP-2 (rhBMP-2) is recognized as one of the most effective osteo-inductive factors that have drawn increasing attention to tissue engineering applications. RhBMP-2 delivered on an absorbable collagen sponge is a US FDA-approved bone graft substitute widely used for bone repair in patients under challenging conditions, such as spinal arthrodesis and open tibial fractures [7,8]. However, natural collagen sponge captures rhBMP-2 only through unstable physical absorption and consumes high doses of rhBMP-2 for clinical therapy, making this treatment costly and often comes with various complications, including ectopic ossification, exaggerated inflammation, and soft tissue hematomas [9,10]. The development of a scaffold with high BMP-2 capture capability and enhanced osteo-inductive activity becomes attractive in research. Some newly reported BMP-2 scaffolds have been assembled with artificial extracellular matrix (ECM) biomaterials including peptide amphiphile nanofibers and 3D electrospun fibers [11,12,13,14], in which BMP-2 was captured with BMP-2-binding motifs such as heparin and affinity peptides to prolong the drug’s retention, and biomaterials were designed to mimic native ECM with an osteo-inductive function. For practical applications, undesirable toxicities from synthetic materials and non-specific molecular interactions remain an issue of concern [15]. Novel scaffolds assembled with safe materials that enable specific capture of BMP-2 are still needed from a clinical perspective.

BMP-2-binding molecules are one key component of BMP-2 scaffolds. The development and application of specific BMP-2-binding molecules have significant therapeutic potential. An earlier BMP-2-specific peptide (NH2-TSPHVPYGGGS-COOH) was reported in 2004 using phage display [16]. Nucleic acid aptamers have antibody-like properties regarding specific molecular recognition of biomolecular surfaces, and may have particular advantages in terms of control of their function, biosafety, ease of assembly, and low selection/manufacturing costs [17].

In the evolution of regenerative biomaterials, aptamers have demonstrated great potential to empower artificial ECM. Hydrogels are important sources for synthesizing artificial ECM due to their mimetic biological and mechanical properties [18]. They are extensively applied in developing regenerative medicine. In the incorporation of aptamers, hydrogel scaffolds exhibit favorable functions to enable specific protein delivery, recruitment of stem cells, promotion of signaling, and molecular biosensing [19,20]. Aptamer-functionalized hydrogels as specific scaffolds have been reported for several growth factors such as VEGF and PDGF [21], but not yet for rhBMP-2. Collagen is a natural hydrogel material that has high biosafety and biocompatibility for in vivo applications. It is able to self-assemble with DNA as fibrous biomaterials to resemble native ECM for drug delivery, wound dressing, cell remodeling, and tissue regeneration [22,23]. Aptamer–collagen fibers were reported recently and demonstrated effects to stimulate the remodeling of angiogenic-like endothelial cells [23]. As inspired, herein, we attempted to integrate DNA aptamer nanotechnology, with DNA-collagen self-assembly to develop an ECM-mimetic scaffold specifically to rhBMP-2 (Figure 1).

In this study, BA1 was identified as the first DNA aptamer specific to BMP-2 with bead-based SELEX. ELONA (enzyme-linked oligonucleotide assay) and EMSA (electrophoretic mobility shift assay) were conducted to identify and characterize the aptamers. The interaction model of the BMP-2-aptamer complex was predicted with HDOCK. Defined DNA aptamers were then applied to assemble ECM-like fibrous scaffolds with native type I collagen. A further investigation of the scaffold efficiency in promoting BMP-2-induced osteogenesis was carried out based on a myoblast cell line, C2C12. The BA1 aptamer–collagen complex has significant potential for downstream therapeutic applications. 

## 2. Results and Discussions

### 2.1. Selected DNA Aptamers Demonstrate Strong Affinity and Specificity to rhBMP-2

Purified His-tagged rhBMP-2 was expressed in *E. coli* (Appendix A). Given the activity in the alkaline phosphatase (ALP) assay (Appendix A), rhBMP-2 was validated to fold into a native conformation after refolding [24], and was therefore used as the target for the aptamer selection. Expressed rhBMP-2s were immobilized on nickel beads with saturation (Appendix A). DNA aptamers against rhBMP-2s were selected using the bead-based SELEX from a library with approximately 10^14^ complexity, in which the ssDNA sequence included one 35 nt random region flanked with two 18 nt fixed primer-binding regions. After 22 rounds of selection, an enrichment of the ssDNA pool to rhBMP-2 was observed in ELONA and EMSA assays (Appendix A). Non-specific aptamers to BMP-2 analogues and histidine tags were excluded by counter selections against rhBMP-3 and His-peptide (Appendix A). In the EMSA assay, the unbound DNA band descended as expected with the increase in BMP-2, suggesting the binding affinity of the DNA pool. However, the bound DNA-rhBMP-2 complexes were not detected. The possible reasons might be as follows: (1) rhBMP-2 precipitated in the native PAGE gel with a pH 8.3 that is near to the isoelectric point (pI) of rhBMP-2 at 8.2 or (2) rhBMP-2 might migrate to the opposite direction due to positive charge. Nevertheless, the enrichment of the ssDNA pool was consistently indicated by the ELONA and EMSA assays. Several enriched pools were then sent for high-throughput sequencing to identify aptamer candidates. 

Representative full sequences showing high copy numbers, high enrichment ratios, and low anticipated free energies from the adenine-rich (BA1–BA5) and non-adenine-rich categories (BNA1-BNA3) were synthesized and characterized (Appendix AA and Table 1). One adenine-rich aptamer, BA1, with the highest copy number from the largest cluster demonstrated a much stronger affinity and specificity to the target, rhBMP-2 (Appendix A). The estimated *K*_D_ value was approximately 7 nM (Table 1 and Figure 1B). Weak non-specific binding to control targets including BMP-3 and BSA was observed (Figure 1C), and negligible affinity to histidine-tagged molecules encompassing His-peptide and His-pfLDH was detected (Figure 1D), indicating that the aptamer binds to rhBMP-2 and probably not to the histidine tag. The dominant adenine-rich sequence, BA2, and non-adenine-rich sequence, BNA2, also showed affinity to rhBMP-2, but with lower affinities than BA1 (Appendix A). Therefore, BA1 was taken forward as the most promising aptamer for further investigation.

Unlike general DNA aptamers, BA1 unusually has a high density of adenine residues. Adenine-rich aptamers were reported in prior studies to target gold surfaces and immobilized nickel ions (Ni^2+^) [25,26]. Since nickel magnetic beads were used as the target support in this study, we examined the affinity of BA1 to nickel beads to confirm its binding specificity. BA1 bound to BMP-2-immobilized and bare nickel beads were collected and amplified with PCR. Appendix A demonstrates that BA1 had slight binding to bare nickel beads and targeted more to BMP-2, as indicated by the excess DNA captured by BMP-2-attached beads. Given the adenine-rich property of BA1, it is possible that BA1 was enriched to both Ni^2+^ and BMP-2 but was more specific to BMP-2. A similar case was reported in a multiplexed SELEX conducted by Alex M Yoshikawa, where a cross-reactive aptamer-SK-1 was identified to simultaneously target several metabolites [27]. Although all nickel beads were saturated by BMP-2 before the selection (Appendix A), the dissociation between Ni^2+^ and histidine tags on BMP-2 caused by the increased salt strength may provide space for ssDNA to target Ni^2+^. Since BA1 was dramatically more specific to BMP-2, it was still used for the following research. Adenine-rich ssDNA has been found to fold distinct secondary structures, like hairpins, duplexes, and bulges, which are likely to develop highly stable tertiary conformations since central adenine bases may stack to decrease the energy barrier and provide a charge transfer channel for the DNA strand [28,29]. To better understand the conformation of BA1 and its possible interactions with rhBMP-2, a structural simulation was carried out next. 

### 2.2. Aptamers May Interact with rhBMP-2 at Heparin-Binding Domains

In the molecular docking analysis, the 3D structure of BA1 was first predicted via RNAComposer [30] based on its secondary structure generated using Mfold [31], which was transformed to a ssDNA 3D structure using Discovery Studio [32] and further refined with HyperChem [33] to stabilize the conformation. Meanwhile, the 3D structure of rhBMP-2 was extracted from the RCSB PDB data bank (ID: 1ES7). Molecular docking was then performed with HDOCK [34] to simulate the interaction model of the BA1-rhBMP-2 complex. BA1 was predicted to adopt hairpin-like structures where the adenine-rich loop is twisted as a knot in the center and terminal nucleotides hybridized to duplexed tails (Figure 2A). The docking model showing the lowest free energy was selected for the BA1-rhBMP-2 complex, in which the interaction interface of BA1 included a partial adenine-rich loop and a few terminal nucleotides, and that of rhBMP-2 was mainly composed of the heparin-binding domain and part of ancillary epitopes close to finger helix grooves [35,36] (Figure 2B). Charge–charge interactions and classical hydrogen bonds might be the dominant mechanisms by which BA1 binds to rhBMP-2 (Figure 2A). 

Aside from BA1, a scrambled sequence was also used as a negative control for the simulation analysis. The molecular docking was conducted in the same way as described above. The scrambled sequence mainly targeted the finger region of rhBMP-2 with its duplex structure (Appendix A), and the heparin-binding domain of rhBMP-2 was not involved in the interaction. The Gibbs free energy change of the scrambled-rhBMP-2 binding model (−255.5 kcal/mol) was significantly lower than that of the BA1-rhBMP-2 complex (−312.3 kcal/mol). Given that Gibbs free energy change (ΔG) is equal to RTInKd, the Kd of BA1 to rhBMP-2 should be much lower than that of scrambled sequence, implying that BA1 was more specific to rhBMP-2 than the scrambled oligonucleotide, which is consistent with our experimental observations. 

Meanwhile, competition binding assays were performed to validate the putative interaction model for BA1 and rhBMP-2. Since the heparin-binding domain on rhBMP-2 was defined as the possible binding interface for BA1 (Figure 2), heparin was used as the binding competitor to BA1 in an experiment to validate the model. Three patterns of binding were investigated: BA1_Heparin indicates pre-treatment of BA1 followed by heparin as BMP-2-binding competitors, Heparin_BA1 indicates pre-treatment of heparin followed by BA1, and Heparin+BA1 indicates simultaneous treatment of BA1 and heparin. ELONA and EMSA were used to evaluate the competitive binding behavior between BA1 and heparin to rhBMP-2. Heparin turned out to be more competitive than BA1 when binding to rhBMP-2 (Figure 3). The sites pre-occupied by BA1 on rhBMP-2 could be slowly replaced with heparin but pre-bound heparin fully inhibited the binding of BA1 to rhBMP-2. This would suggest that BA1 and heparin likely share the same heparin-binding site on rhBMP-2. Consistent results were observed when using the heparin-like molecule polyphosphate [37] as a binding competitor (Figure 3). Collectively, our data were consistent with BA1 binding at the heparin-binding domain. The underlying mechanism for this interaction mode could be due to the strong electrostatic interactions favored by the positive charge at heparin-binding epitopes and the negative charge at phosphate linkage bonds in DNA nucleotides. This would also be consistent with data showing markedly reduced affinity of BA1 to rhBMP-2 in a high-salt binding buffer (Appendix A). 

### 2.3. Assembly of Aptamer-Modified Collagen Fibers as rhBMP-2′s Scaffolds

Aptamer–collagen fibers were assembled using a formulation of 500 nM ssDNA and 10% type I collagen (0.3 mg/mL) (Figure 4). Both the BA1 aptamer and a scrambled sequence of the same length were co-localized with collagen in scaffolds adopting indiscriminate morphologies (Figure 5). The self-assembly of ssDNA-collagen fibers was likely to be induced by the hydrogen associations between specific donors with strong dipole moment in collagen and negatively-charged phosphate acceptors in DNA sequences [38]. The morphology of the fiber was mainly determined by the length, instead of the nucleotide or the strandedness, of the DNA strand [39], which could explain the observation of similar patterns in both BA1 and scrambled DNA fibers. However, fibers derived from different formulations varied in architecture, size, and density (Appendix A). Moreover, 5% collagen was not enough to induce the formation of fibers. In addition, a lower volume fraction of collagen resulted in thinner and shorter fibers, while a higher proportion led to more compacted fibrous bundles. More collagens may allow more DNA molecules to be displayed, leading to the aggregation of larger bundles. The aggregation cannot continue when the volume fraction of collagen reaches 50%. Highly dense complexes might shield the binding domain at the surface.

To examine whether aptamers maintain binding affinities to rhBMP-2 in fibrous scaffolds, we performed an ELISA-like assay where rhBMP-2 solutions were added into streptavidin-coated plate wells immobilized with biotinylated aptamer–collagen fibers to allow for molecular interactions. The aptamer BA1 retained its specific binding to BMP-2 when it was assembled with type I collagen with an estimated *K*_D_ at 3.5 nM (Figure 6A). However, the binding capacity of BA1 in fibers decreased relatively to free BA1 since the plate wells coated by BA1-collagen fibers were observed to be saturated with less BMP-2 than with free BA1. The possible reason could be that some binding sites of BA1 in fibers may be encapsulated or blocked by collagen molecules to impede the binding to BMP-2. In addition, it is observed that the affinity of BA1 in fibers kept decreasing along with the increase in collagen’s volume fraction (Appendix A), which might be attributed to the reduced functional aptamers in fibers showcasing aggregates and lowered density. Fibers with smaller diameters have been reported to have larger surface-area-to-volume ratios to enable higher loading capacities than thicker fibers [40]. Fibers assembled with 10% collagen were thinner and more dispersed than those fabricated with high volume fractions of collagen (30% and 50%), thereby contributing to the larger surface area and sample loading capability. Among all testing fibers, the one formulated with 500 nM ssDNA and 10% collagen (0.3 mg/mL) demonstrated the strongest binding to rhBMP-2. This formulation was therefore used in later experiments. 

The stability of aptamers in fibers was also investigated to enable a reliable functional assessment of BMP-2 in vitro. Since the following functional experiments all lasted less than 24 h or changed fresh medium every 24 h, the stability of free or collagen interlinked BA1 and scramble sequences incubated with cell culture medium at 37 °C within 24 h were analyzed. BA1 and scramble DNA maintained stability for at least 24 h in the cell culture medium, and they demonstrated similar stability in collagen fibers (Figure 6B–E). The assembly of the DNA-collagen complex was reported to stabilize dsDNA against nucleases by stabilizing its hydration shell in water [38]. However, the improvement of ssDNA stability was not observed in our study. The lack of sufficient hydrogen bonds in ssDNA between base pairs to interact with surrounding water molecules may lead to unimproved stability. 

### 2.4. Aptamer-Enabled Scaffolds Tended to Enhance rhBMP-2-Mediated Osteogenesis 

With well-characterized aptamer–collagen fibrous scaffolds, we then evaluated their effects in promoting the osteo-inductive activity of rhBMP-2 on the myoblast cell line, C2C12. C2C12 has been used as a cell model to investigate the mechanism by which BMP-2 promotes osteoblastic differentiation [41]. 

A comparison was made first to assess the abilities of free BMP-2 and fiber-loaded BMP-2 to induce the expression of alkaline phosphatase (ALP), a representative marker for the osteoblastic differentiation of C2C12 [42]. Based on a dose–response pilot assay, 200 ng/mL of *E. coli*-expressed BMP-2 was found to be an ALP-inducible threshold and was therefore used as the treatment condition. ALP expression levels were increased with BMP-2 loaded on aptamer–collagen fibers, and a significant improvement was observed when compared with the use of scramble-assembled fibers (Figure 7). As expected, the bare fiber did not potentiate the ALP expression in cells in the absence of BMP-2. Furthermore, only the full assembly could promote ALP expression. Either aptamer or collagen alone was less effective to enhance BMP-2 than the integrated fibers.

We next investigated the impact of aptamer–collagen fibers on BMP-2-stimulated mineralization [43]. The formation of minerals is driven by ALP in bone, the level of which reflects the osteo-inductive capacity of BMP-2-loaded scaffolds. We first examined the mineralization on DNA-collagen fibers since they were observed to retain calcium deposits as collagen fibrils in native ECM [22]. Herein, the formation of calcium was induced using a mineralization solution (25 mM NaCl, 8 mM Na_2_HPO_4,_ and 15 mM CaCl_2_). After overnight incubation, DNA-collagen fibers, especially the aptamer-interlinked fibers, captured a remarkable amount of calcium deposits (Appendix A), suggesting that the aptamer-functionalized fiber holds promise to be developed as osteo-inductive coating materials as suggested by Bryan D James [22]. The interaction between DNA fibers and the calcium solution occurs between the phosphate backbone of DNA and surface calcium atoms [22]. The observation in our study might be caused by the fact that the conformation of BA1 enabled it to expose more phosphate backbone area than the scramble sequence to calcium phosphate to facilitate the mineralization process. Meanwhile, the mineralization assay was also carried out on C2C12 cells where calcium deposits were induced by BMP-2. BMP-2 loaded on the aptamer-functionalized fibers demonstrated enhanced efficiency to trigger mineralization in comparison to free BMP-2 (Appendix A). However, the improvement was not dramatic when compared to that on the scramble-assembled fibers. One possible explanation might be due to the short induction period. Aptamer–collagen fibers with enhanced stability should be developed to enable a long-term study (e.g., 21 days) of cellular mineralization. 

In addition, cell adhesion and wound healing assays were carried out to explore the possibility of using aptamer-functionalized fibers for bone healing applications. Bone regeneration is initialized by cell adhesion [44]. ECM could promote the attachment of osteoblast-like cells, followed by cell migration, proliferation, and differentiation to induce new bone formation [45]. As ECM-like materials, aptamer–collagen fibers were able to induce cell adhesion as expected (Figure 8). C2C12 cells treated with BMP-2 in the presence of collagen or BA1-incorporated collagen fibers were well attached and adopted dendritic and elongated structures. In contrast, BMP-2 untreated or on DNA-coated plate wells were less adhesive according to the limited stretched morphologies. Relative to free BMP-2, BMP-2 loaded on fibers exhibited enhanced cell adhesion in light of the increased cell area and cell number. This indicates the potency of aptamer–collagen fibers to enhance BMP-2-induced cell adhesion. In the comparison with scrambled DNA-formulated fibers, the total area of cells on aptamer-assembled fibers did not show a significant increment. Cells grown with aptamer-functionalized fibers grew too fast and reached high confluence within 10 h. Most of these cells were squeezed and could not fully stretch, which may lead to the underestimated cell area. However, given the significantly increased cell number, we could still conclude that aptamers exhibited specific advantages in scaffolding fibers to enhance cell adhesion induced by rhBMP-2.

We further investigated wound healing to measure tissue regeneration at the wound margin layer [46]. It is observed that BMP-2 accelerated the wound healing of C2C12 cells in comparison to the untreated group (Figure 9), which was consistent with the finding that BMP-2 induces cell migration [47]. When displayed with aptamer–collagen fibers, cellular wounds healed in a faster manner at 4 h and 12 h, suggesting that this fibrous scaffold promotes BMP-2-induced wound healing. In the comparison with scrambled DNA-assembled fibers, the augmented wound healing from aptamer-functionalized fibers was observed at 12 h but not at early stages (4 h). In total, 4 h of culture may be too short for live cells to accumulate BMP-2 at a functionally significant level to initiate the signaling pathways for cell renewal. Similar phenomena were observed in Amira Seltana’s study [48]. Given the statistically significant improvement at 12 h, we proposed that BA1 aptamers at collagen fibers benefited BMP-2 more over scramble sequences to stimulate wound healing progression.

A clear implication from the above data is that aptamer BA1-functionalized collagen fibers promote the osteo-inductive activity of BMP-2 in an in vitro context. Aptamer–collagen fibers are not new in material science. A recent study by Bryan D James used VEGFR2-specific DNA aptamers to assemble collagen fibers that elicited impressive effects to induce angiogenic-like cell remodeling for regenerative medicine applications [23]. Similarly, the fibrous scaffold in this study was assembled with BMP-2-specific DNA aptamers and type I collagen. By displaying growth factors (BMP-2) at the surface, the fibers may mimic the function of the extracellular matrix (ECM) given their analogous architecture and composition. ECM plays a crucial role in stimulating the formation of new bone by providing essential cytokines, mechanical support, and osteoblast interactions [49]. The observations in our research are consistent with the role of ECM regarding enhanced osteoblastic differentiation, cell adhesion, and wound healing. Our observations are also in line with Sungsoo S Lee’s study where ECM-like nanofibrous scaffolds self-assembled with BMP-2-binding peptide amphiphiles, thereby enhancing BMP-2 signaling and bone fusion in a rat posterolateral lumbar intertransverse spinal model [13]. In comparison with most synthetic scaffolds for BMP-2, the prominent advantages of our scaffolds include (1) ease of assembly and modification, (2) specific capture of BMP-2, and (3) low toxicity from raw materials. For proceeding to in vivo studies, the aptamer–collagen fibers should be encapsulated in 3D matrix biomaterials such as agarose to enable controllable drug release. Moreover, the fibrous scaffold requires further refinement to improve nuclease resistance and osteo-inductive potency. DNA aptamers and collagen are highly biodegradable in vivo, and the combination of two materials was observed to stabilize fibers in the cellular milieu [38]. To achieve long-lasting efficacy in vivo, fibers could be further stabilized by modifying aptamers with 2′-F, 2′-OMe or phosphorothioate bonds [50] and constructing composite hydrogels with more resilient biomaterials such as agarose [51]. In addition, to improve the potency for bone regeneration, some strategies that could be adopted include (1) an optimization of fiber formulation to generate unified fibers with efficient functional morphologies, (2) the incorporation of additional growth factors like VEGF into fibers to enable synergistic regenerative effects [52], and (3) the integration of cell-binding moieties to recruit bone marrow-derived mesenchymal stem cells (BMSCs) or osteoblast lineage (MC3T3-E1 cells, etc.) to achieve cell-free in situ bone regeneration [53,54]. 

## 3. Materials and Methods

### 3.1. Materials

C2C12 myoblast cell line was gifted by Prof. Zhongjun Zhou from the School of Biomedical Sciences at the University of Hong Kong. Dulbecco’s Modified Eagle Medium (DMEM), 0.25% Trypsin (EDTA), and fetal bovine serum (FBS) were purchased from Thermo Fisher Scientific (Waltham, MA, USA). Oligonucleotides were ordered from Integrated DNA Technologies (Coralville, IA, USA) and Hippobio (Huzhou, China). Primary rhBMP-2 antibody and anti-mouse secondary antibody were obtained from Abcam (Cambridge, UK). Ni-NTA agarose magnetic beads and QIAquick PCR Purification Kit were both from Qiagen (Hilden, Germany). Pwo DNA polymerase was obtained from Roche (Basel, Switzerland). The dNTP set and MyOne C1 streptavidin magnetic beads were obtained from Thermo Fisher Scientific (Waltham, MA, USA).

### 3.2. Expression, Renaturation, and Purification of His-Tagged rhBMP-2 Targets

cDNA encoding the mature region of rhBMP-2 (Gene ID: 650) was synthesized by GenScript (Piscataway, NJ, USA), and ligated into pET-15b (GenScript, USA) between NdeI and BamHI sites to construct rhBMP-2 expression plasmid, followed by the transformation into BL21 (DE3) pLysS-competent cells for isopropyl β-D-1-thiogalactopyranoside (IPTG)-induced protein expression. Bacteria were cultured in LB broth medium with 100 μg/mL of ampicillin at 37 °C for 1–2 h until OD600 reached 0.5–0.8. Then, 0.5 mM IPTG was added and followed by 4 h of incubation at 37 °C with shaking at 220 rpm to express proteins. Cell pellets were harvested by centrifugation at 4000× *g* for 45 min at 4 °C. The resultant cell pellets were washed twice with 30 mL cold PBS buffer. rhBMP-2-containing inclusion bodies (IBs) were extracted by the buffer containing 6 M Gdn-HCl, 0.1 M Tris (pH 8.5), 1 mM EDTA, and 0.1 M DTT with overnight incubation at RT. The suspension was collected as soluble Ibs and exchanged to the buffer with 5 M Gdn-HCl and 50 mM MES (pH 5.0). rhBMP-2 in the soluble Ibs was refolded using a rapid dilution in the buffer containing 0.5 M L-arginine, 0.1 mM GSSG, 1 mM GSH, 20 mM Tris-HCl (pH 8.5), and 5% glycerol at a ratio of 1:25 (*v*/*v*). Refolded proteins were dialyzed to the binding buffer with 4 M urea, 20 mM Tris-HCl (pH 8.0), and 150 mM, and applied to the HiTrap Heparin column (Cytiva, Marlborough, MA, USA) for protein purification. A gradual elution program was carried out on the ÄKTA Pure system (Cytiva, USA) in the buffer containing 4 M urea, 20 mM Tris-HCl (pH 8.0), and NaCl (150 mM–1.5 M). All fractions of interest were collected and analyzed with 12% SDS-PAGE gel. rhBMP-2 dimers were pooled together and exchanged to the buffer containing 50 mM MES, pH 5.0. The final products were concentrated at 1 mg/mL and stored at −80 °C until further use.

### 3.3. Aptamer SELEX

The single-strand DNA (ssDNA) contained a 35 nt random region flanked with two 18 nt sequences for PCR primer annealing and amplification (5′-CGTACGGTCGACGCTAGC-[N35]-CACGTGGAGCTCGGATCC–3′). Initially, 1 nmole of ssDNA library was diluted in 100 μL binding buffer (20 mM HEPES, 2 mM MgCl_2_, 2 mM CaCl_2_, 2 mM KCl, 150 mM NaCl, pH 7.4, 0.05% Tween-20) and then heated at 95 °C and cooled down on the ice for 5 min, followed by a slow recovery to room temperature for 10 min. His-tagged rhBMP-2 expressed from *E*. *coli* was immobilized to nickel magnetic beads with 1 h incubation at RT in the protein binding buffer with 20 mM imidazole, 20 mM HEPES, 2 mM MgCl_2_, 2 mM CaCl_2_, 2 mM KCl, 150 mM NaCl, pH 7.4, and 0.05% Tween-20. The rhBMP-2-beads were incubated with the library at 37 °C for 30 min with slow rotation. Unbound ssDNA sequences were removed by being washes 3–6 times with the washing buffer (20 mM HEPES, 2 mM MgCl_2_, 2 mM CaCl_2_, 2 mM KCl, 0.15 mM/0.5 M/1 M NaCl, pH 7.4, 0.05% Tween-20), and the remaining species were resuspended in 20 μL PCR-grade water to carry out PCR amplification in the PCR mixture (primers, dNTP and PWO polymerase). Afterwards, PCR products were purified with the QIAquick PCR purification kit (Qiagen, Germany) and conjugated to MyOne streptavidin T1 beads (ThermoFisher, Waltham, MA, USA) according to the manufacturer’s instructions, and ssDNAs of interest were eluted with 0.1 M NaOH for 5 min to obtain the ssDNA library for the next round of selection. The concentration of each ssDNA pool was measured with Nanodrop 1100 (ThermoFisher, USA) at 260 nm. After 22 rounds of selection, the enriched ssDNA pools were sent for high-throughput sequencing (HTS).

Over the 22 rounds of selection, the selection stringency was gradually increased as follows: (1) the inputs of protein targets and incubation time were reduced; (2) the amount of ssDNA library decreased from 1 nmol to 50 pmole; (3) the volume of binding buffer increased from 0.2 mL to 1 mL from Round 2; (4) counter selections against beads were included in Rounds 2, 9, 16, 18–20, and counter selections against negative targets (rhBMP-3 and His-peptide) were added in Rounds 7, 12, 16–19; (5) 0.5 mg/mL tRNA was added starting from Round 14; and (6) incubation time was reduced from 30 min to 15 min after Round 10 and to 10 min after Round 19.

### 3.4. High-Throughput Sequencing (HTS)

After the selection, several enriched ssDNA pools with the initial library (Round 0, Round 1, Round 5, Round 10, Round 15, and Round 22) were sent for HTS sequencing. They were amplified with unmodified internal primers flanked by unique barcodes. The yielded PCR products were purified with 4% agarose gel and extracted with a QIAquick gel extraction kit (Qiagen, Germany). ssDNA pools were mixed in equal amounts and further amplified using external primers attached with Illumina-specific sequencing adaptors. After gel purification, 30 ng of PCR sample was sent to Novogene (Beijing, China) for HTS analysis. Sequencing data were analyzed with FASTAptamer (https://burkelab.missouri.edu/fastaptamer.html (accessed on 16 December 2020) based on the manufacturer’s instructions.

### 3.5. EMSA

EMSA was conducted by following the protocol in our previous research [55]. In brief, the protein was serially diluted in the binding buffer (20 mM HEPES, 2 mM MgCl_2_, 2 mM CaCl_2_, 2 mM KCl, 150 mM NaCl, pH 7.4). Aptamer sequences were mixed with the protein solution to a final concentration of 25 nM, followed by incubation at 37 °C for 1 h. The resultant complexes were loaded on 12% native PAGE polyacrylamide gels for 30 min electrophoresis at 4 °C. Gels were then visualized with SYBR gold nucleic acid staining (ThermoFisher, USA) on the ChemiDoc Imaging System (BioRad, Hercules, CA, USA).

### 3.6. ELONA

In total, 100 μL of 100 ng of rhBMP-2 was immobilized on the pre-blocked HisSorb plate (Qiagen, Germany) in the PBST buffer (0.05% Tween-20). Unbound rhBMP-2 were removed by being washed three times with 200 μL PBST buffer. Biotinylated DNA aptamers were heated at 95 °C for 5 min and cooled down on the ice for 10 min. Then, 100 μL of annealed aptamer sequences were incubated with immobilized rhBMP-2 for 1 h at 37 °C. After removing unbound aptamers, 100 μL of HRP-conjugated streptavidin (1:100,000 dilution) solution was added and incubated for 30 min at RT. Following three washes with 200 μL of PBST, 100 μL of TMB solution (ThermoFisher, USA) was added and incubated for 5–15 min at RT. The reaction was stopped by 0.16 M H_2_SO_4_, and the absorbance was measured at 450 nm by the Varioskan LUX Multimode Microplate Reader (ThermoFisher, USA). In the binding competition assay, the HisSorb plate was immobilized with the same amount of rhBMP-2. Moreover, a 100 nM aptamer was added before, with, and after the heparin/polyphosphate serial solution (0–10,000 nM) to allow 1 h of incubation with immobilized rhBMP-2. The bound aptamer was determined with the same method as described above.

### 3.7. Molecular Docking

The three-dimensional structure of rhBMP-2 dimer was retrieved from the PDB file, ID: IES7. The 2D structure of the DNA aptamer was predicted via Mfold and converted to 3D conformation using RNA composer (https://rnacomposer.cs.put.poznan.pl/ (accessed on 16 June 2021). Discovery Studio (Dassault Systemes BIOVIA, San Diego, CA, USA) was then used to recover the DNA aptamer sequence by replacing all uracil with thymidine and ribose with deoxyribose. Afterwards, the 3D structure of the aptamer was refined with HyperChem (Hypercube Inc., Gainesville, FL, USA) using structural energy minimization to obtain a stable structure for molecular docking. The PDB files of rhBMP-2 and the refined aptamer were inputted to the HDOCK server (http://hdock.phys.hust.edu.cn/ (accessed on 18 August 2021)), which sampled all possible molecular interaction modes using a knowledge-based search algorithm and evaluated them with an energy function called ITScorePP. The top 10 binding models from HDOCK were analyzed with Discovery Studio to visualize the interaction interface.

### 3.8. Assembly of Aptamer–Collagen Fibrous Scaffolds

Aptamer–collagen fibers were assembled similarly as reported by Bryan D James [23]. In brief, amine-capped aptamer and scramble ssDNA sequences were diluted in sterile water to the working concentration of 1 μM. Type I collagen solution in 0.02 M acetic acid was diluted to 0.3 mg/mL using sterile water. ssDNA solution was mixed with type I collagen with different volume fractions (0%, 5%, 10%, 20%, 30%, and 50%), followed by overnight incubation at room temperature to allow for the spontaneous assembly of ssDNA-collagen fibers.

### 3.9. Surface Immobilization of Aptamer–Collagen Fibers

Fibers were immobilized on well plates using the sulfo-SANPAH heterobifunctional crosslinker. In total, 20 μM of sterile crosslinker aqueous solution was added to well plates and exposed to ultraviolet light at 365 nm for 10 min to activate the nitrophenyl azide group for binding to the polystyrene surface. Afterwards, the plate wells were rinsed with sterile water three times and incubated with scaffold solutions overnight at room temperature, followed by three rounds of washing with sterile water to remove unbound fibers. The plate was stored at 4 °C until further use. Fibers were photographed with Olympus CKX53 Inverted Microscope (Olympus, Tokyo, Japan).

### 3.10. Elemental Mapping of Aptamer–Collagen Scaffolds

ssDNA sequences in this assay were modified with Cy5 fluorophore, and type I collagen was visualized with Alexa Fluor^@^ 488-conjugated antibody. Upon fiber assembly, fibers were incubated with type I collagen antibody overnight at 4 °C, and excess antibodies were removed by being washed three times using PBS containing 1% BSA. Fibers were then visualized with Olympus IX71 inverted fluorescence microscope (Olympus, Japan).

### 3.11. Enzyme-Linked Immunosorbent Assay (ELISA)

Biotinylated aptamers and their self-assembled collagen fibers were immobilized on streptavidin-coated 96-well plates (ThermoFisher, USA). In total, 100 μL of 0–100 ng rhBMP-2 solutions in the ELISA working buffer was added and incubated with fibers at 37 °C for 1 h. Afterwards, the plate wells were washed with PBST buffer three times. Then, 100 μL rhBMP-2 primary antibody (1:2000) and HRP-conjugated secondary antibody (1:10,000) were added successively to allow 1 h of incubation at room temperature. After three rounds of washing, 100 μL TMB solution (ThermoFisher, USA) was added for 5–10 min of incubation. The reaction was stopped with 100 μL of 0.16 M H_2_SO_4_. The absorbance at 450 nm was measured with POLARstar Omega Plate Reader (BMG LABTECH, Ortenberg, Germany).

### 3.12. Alkaline Phosphatase (ALP) Assay

Aptamer–collagen fibers were immobilized on 96-well cell culture plates using sulfo-SAHPAH crosslinkers. RhBMP-2 was coated on fibers by 1 h incubation at 37 °C with 5% CO_2_. C2C12 cells were then seeded on treated plate wells at a density of 10^4^ cells/well. Cell medium was replaced with 200 μg/mL rhBMP-2 on a daily basis for three days. Afterwards, cells were washed with PBS buffer three times and lysed with 55 μL lysis buffer (1 mM MgCl_2_, 1 mM ZnCl_2_, 1% (*v*/*v*) Igepal CA-630, 0.1 M glycine, pH 9.6) for 1 h at room temperature. After brief centrifugation, 50 μL pNpp substrate (2 mg/mL) was added for 60 min incubation at 37 °C. The absorbance at 405 nm from ALP-pNpp product was determined with POLARstar Omega Plate Reader (BMG LABTECH, Ortenberg, Germany). Protein concentrations of cell lysates were measured with a BCA protein assay kit (ThermoFisher, USA) following the manufacturer’s instructions. The ALP level was normalized as ALP absorbance/total protein amount (mg).

### 3.13. Cell Adhesion Assay

C2C12 cells were seeded on 24-well plates immobilized with rhBMP-2-loaded fibrous scaffolds at a density of 2 × 10^4^ cells/well. After 10 h of culture in the BMP-2-supplemented DMEM medium, cells were fixed and permeabilized with 4% formaldehyde and 0.1% Triton X-100, followed by the nucleus staining with Hoechst (ThermoFisher, USA) for 5 min and cytoskeleton staining with phalloidin-iFluor 488 (Abcam, UK) for 1 h at room temperature. After three rounds of washing with PBS buffer, cells were observed under Olympus IX71 inverted fluorescence microscope (Olympus, Japan). Cell area and cell density were determined with Image J (National Institutes of Health and University of Wisconsin, USA) and further quantified using Origin (OriginLab, Corporation, Northampton, MA, USA).

### 3.14. Wound Healing Assay

C2C12 cells were seeded on 24-well plates immobilized with rhBMP-2-loaded fibrous scaffolds at a density of 2 × 10^4^ cells/well. Cell wounds were created with a physical scratch in the horizontal center. Cells were washed twice with PBS buffer and cultured with a BMP-2-containing DMEM medium with 2% FBS. The wound closure was monitored at different time points, including 0-h, 4-h, 12-h, and 24-h. Images were taken with Olympus CKX53 inverted microscope (Olympus, Japan) and analyzed using Image J (National Institutes of Health and University of Wisconsin, USA).

### 3.15. Calcium Nodule Detection

C2C12 cells were seeded on 96-well plates immobilized with rhBMP-2-loaded fibrous scaffolds at a density of 10^4^ cells/well. Cell medium with 200 μg/mL of rhBMP-2 was changed daily up to another 14 days. Afterwards, cells were washed with PBS buffer three times and fixed and permeabilized with ice-cold 70% ethanol for 1 h. Cells were then washed with excess sterile water and stained with Alizarin Red staining (AR-S) solution (Sigma-Aldrich, St. Louis, MO, USA) for 1 h at room temperature with gentle agitation. Then, AR-S was removed, and cells were washed with sterile water and photographed under Olympus CKX53 inverted microscope (Olympus, Japan). The stained calcium nodules were extracted with 10% (*w*/*v*) cetylpyridinium chloride (CPC) in 10 mM sodium phosphate, pH 7.0, for 1 h at room temperature. The absorbance of dissolved calcium nodules was measured at 560 nm on a POLARstar Omega Plate Reader (BMG LABTECH, Ortenberg, Germany).

### 3.16. Statistics and Data Analysis

Statistical analyses were carried out on Origin 2021b (OriginLab Corporation, Northampton, MA, USA) and GraphPad Prism 8.3.0 (GraphPad Software, San Diego, CA, USA). Comparisons between the two treatment groups were made using unpaired t-tests. The alpha level is lower than 0.05 denotes the statistical significance. EC_50_ and K_D_ values in this study were computed under the non-linear fit models using the function of DoseResp (EC_50_ = 10^LOGx0^, LOGx0 = center of y axis) for Growth/Sigmoidal in Origin 2021 and Hyperbola (y = $\frac{Bmax\cdot x}{kD+x}$, x is the ligand concentration, y is the specific binding, and B_max_ is maximum binding; K_D_ = x with half binding at equilibrium) for one site-specific binding in GraphPad Prism 8, respectively.

## 4. Conclusions

This study demonstrated the evolution of a novel DNA aptamer, BA1, which exhibited high affinity and specificity to the growth factor applied for bone regeneration, rhBMP-2. Through rigorous counter selections, BA1 markedly discriminated rhBMP-2 and its analog rhBMP-3 to achieve selective drug capture as functional motifs on scaffolds. In addition, molecular docking analysis was performed to better understand the molecular interactions of the BA1-BMP-2 complex. The putative interaction model revealed that BA1 may adopt hairpin-like conformations and bind to rhBMP-2 mainly at the heparin-binding domain, which was consistent with the experimental binding competition assays with heparin and polyphosphate. BA1 was then applied to assemble fibrous scaffolds with type I collagen, which could promote the osteogenesis ability of rhBMP-2, as observed in the assays measuring ALP, mineralization, cell adhesion, and wound healing.

This research, however, is subject to several limitations. First, the BA1 aptamer was selected against rhBMP-2 with histidine tags. Histidine is positively charged and may contribute to the binding affinity of the negatively charged DNA aptamer. Although we excluded this possibility by conducting specificity assays with other histidine-tagged targets (His-peptide, His-pfLDH), more direct evidence is needed to confirm the binding capability of BA1 when using BA1 to target unmodified rhBMP-2. Second, the DNA-interlinked collagen fibers were not well-characterized and normalized. This may introduce some bias when identifying the difference between collagen fibers formed by aptamers and scramble sequences. Transmission electron microscopy (TEM) imaging and X-ray diffraction can be used to further determine and normalize the morphologies and sizes of DNA-collagen fibers to enable us to conduct more accurate comparable studies. Nevertheless, our study provides a paradigm of developing aptamer-functionalized collagen fibers for enhancing BMP-2-induced osteogenesis. This could be beneficial for developing BMP-2-mediated regenerative medicine for bone regeneration applications.

## Data Availability

The data presented in this study are available in article and Appendix A.

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
