# Peer review of "The Evolution and Application of a Novel DNA Aptamer Targeting Bone Morphogenetic Protein 2 for Bone Regeneration"

_molecules, 2024, doi:10.3390/molecules29061243_

Round 1
Reviewer 1 Report
Comments and Suggestions for Authors
Liu et al describe the in vitro selection of DNA aptamers to the recombinant human bone morphogenetic protein 2 and the application of the aptamer for inducing bone growth. The work is novel as this is the first reported specific aptamer for the target. Furthermore, the work may open exciting therapeutic avenues for bone damage. The work will be of interest to readers and should be published after minor changes. Below are comments in support of the article as well as some required changes.
There are a few too many acronyms in the title, can you make it more general? For instance, I didn’t know what rhBMP-2 was and even “ECM” might not be obvious to a chemist, engineer, etc (many researchers who could be interested in aptamers).
After 22 rounds of a selection, a DNA aptamer emerged with high affinity in the nanomolar range. Is the sequencing data available? If not, please at least elaborate on the number of reads that were used in the Fastaptamer analysis.
Importantly, the authors used two methods to ensure high affinity binding. This is a requirement from the Aptamer Society and thus is very welcomed here. The aptamer was also assessed for specificity and notably the authors confirmed that binding was not due to the His tag used to isolate the expressed protein. Excellent work!
Next, the authors assessed the binding domain of the aptamer. To do this, they combined molecular docking and competition assays with known inhibitors/binders. From this work, they show that the aptamer interacts with the heparin-binding site. With these details in hand, aptamer-collagen fibers were assembled to serve like an extracellular matrix-like scaffold. The fibers were tested for their stability and the maintained function of the aptamers. All work was compared to a “scrambled” sequence with the same length. Despite that these results were not as exciting as hoped, the authors carefully explain the potential findings. However, I could not find the scrambled sequence, nor the aptamer sequence. Please add this especially since there is a comment in the paper that says “Samples of the oligos are available from the companies listed” – but how would someone know what to order?
Finally, the scaffolds were assessed to determine whether they could promote osteoinductive activity on the myoblast cell line C2C12. For the results in figure 6, 7, 8, please write which statistical test was used to generate these P values, and if it was 3 separate experiments or just technical replicates, directly in the figure title. Can you also please put a scale bar on figure 7 and 8?
Some of the supporting figures are called out in the main manuscript and seem very critical to the story such as Figure S10A, S11. I would therefore like to see them in the main text, perhaps as a composite figure of a main text figure.
Finally, the first 2 sections of the results are easy to read. After that, it is a bit hard to follow the story. ne main thing that can be changed and will drastically help the flow of the story is to put the figures in brackets. This was done well in the first sections – for example “Purified His-tagged rhBMP-2 was expressed in E. coli (Figure S1A).” However, in most of the results section, the sentence structure focuses on the figures and therefore it’s hard to connect the thoughts. Examples include “Figure S6 demonstrates that BA1 had slight…”; and “As shown in Figure S9, 219 fibers varied in architecture, size, and density in formulations with different collagen vol-220 ume fractions”. Please change all of these (and many more) so that the Figure and number are listed at the end of the statement, and in a bracket as to not distract from the main point. It will keep the ideas/logic flowing much more smoothly.
Author Response
Reviewer 1:
"Liu et al describe the in vitro selection of DNA aptamers to the recombinant human bone morphogenetic protein 2 and the application of the aptamer for inducing bone growth. The work is novel as this is the first reported specific aptamer for the target. Furthermore, the work may open exciting therapeutic avenues for bone damage. The work will be of interest to readers and should be published after minor changes. Below are comments in support of the article as well as some required changes.
There are a few too many acronyms in the title, can you make it more general? For instance, I didn’t know what rhBMP-2 was and even “ECM” might not be obvious to a chemist, engineer, etc (many researchers who could be interested in aptamers).
Response: Thanks for your great suggestion. We changed a more general title to “Evolution and Application of a Novel DNA Aptamer Targeting Bone Morphogenetic Protein 2 for Bone Regeneration”.
After 22 rounds of selection, a DNA aptamer emerged with high affinity in the nanomolar range. Is the sequencing data available? If not, please at least elaborate on the number of reads that were used in the Fastaptamer analysis.
Response: No problem. The Fastaptamer analysis results are shown in Figure S5A and we added the number of reads of all test oligos in the table as suggested. Besides that, RPM (reads per million) is also included which is frequently used to count up the reads of oligos in the sequencing pool. It divides the read counts by the “per million” scaling factor and normalizes for sequencing depth. The principle of selecting aptamer candidates for the test in this study is based on the copy number, putative free energy, and enrichment degree of the oligo. For more details of the sequencing results please kindly refer to the text below in the manuscript,
“…Representative full sequences showing high copy numbers, high enrichment ratio, and low anticipated free energies from the adenine-rich (BA1-BA5) and non-adenine-rich categories (BNA1-BNA3) were synthesized and characterized (Figure S5A and Table 1). One adenine-rich aptamer, BA1, with the highest copy number from the largest cluster demonstrated a much stronger affinity and specificity to the target, rhBMP-2 (Figure S5)…”
Importantly, the authors used two methods to ensure high-affinity binding. This is a requirement from the Aptamer Society and thus is very welcomed here. The aptamer was also assessed for specificity and notably, the authors confirmed that binding was not due to the His tag used to isolate the expressed protein. Excellent work!
Next, the authors assessed the binding domain of the aptamer. To do this, they combined molecular docking and competition assays with known inhibitors/binders. From this work, they show that the aptamer interacts with the heparin-binding site. With these details in hand, aptamer-collagen fibers were assembled to serve like an extracellular matrix-like scaffold. The fibers were tested for their stability and the maintained function of the aptamers. All work was compared to a “scrambled” sequence with the same length. Despite that these results were not as exciting as hoped, the authors carefully explain the potential findings. However, I could not find the scrambled sequence, nor the aptamer sequence. Please add this especially since there is a comment in the paper that says “Samples of the oligos are available from the companies listed” – but how would someone know what to order?
Response: Thanks for your kind reminder. The sequences of aptamer and scramble oligos are shown in Table 1. BA5 was selected as the scramble oligo for comparison in this research since it has the same base number and is adenine-rich as the aptamer oligo.
Finally, the scaffolds were assessed to determine whether they could promote osteoinductive activity on the myoblast cell line C2C12. For the results in figure 6, 7, 8, please write which statistical test was used to generate these P values, and if it was 3 separate experiments or just technical replicates, directly in the figure title. Can you also please put a scale bar on figure 7 and 8?
Response: No problem. We have modified these figures as you suggested. Figure numbers 6-8 were changed to 7-9. The statistical test used was the unpaired t-test since either two treatment groups were independent. Statistic results were from three separate experiments. The scale bar for Figure 8 and 9 were also added. Please kindly refer to Figure 6-8 and their legends in the manuscript for more details.
Some of the supporting figures are called out in the main manuscript and seem very critical to the story such as Figure S10A, S11. I would therefore like to see them in the main text, perhaps as a composite figure of a main text figure.
Response: We agree with your opinion. As suggested, Figure S10A and S11 were composed as Figure 6 and it was moved to the main text.
Finally, the first 2 sections of the results are easy to read. After that, it is a bit hard to follow the story. The main thing that can be changed and will drastically help the flow of the story is to put the figures in brackets. This was done well in the first sections – for example “Purified His-tagged rhBMP-2 was expressed in E. coli (Figure S1A).” However, in most of the results section, the sentence structure focuses on the figures and therefore it’s hard to connect the thoughts. Examples include “Figure S6 demonstrates that BA1 had slight…”; and “As shown in Figure S9, 219 fibers varied in architecture, size, and density in formulations with different collagen volume fractions”. Please change all of these (and many more) so that the Figure and number are listed at the end of the statement, and in a bracket as to not distract from the main point. It will keep the ideas/logic flowing much more smoothly.
Response: Thank you for this great suggestion. Accordingly, we improved the writing as you proposed throughout the manuscript. The sentence structure was changed by putting the statement ahead and moving the figure and number to the end in a bracket, such as “…However, fibers derived from different formulations varied in architecture, size, and density (Figure S9)…”.

Reviewer 2 Report
Comments and Suggestions for Authors
The research by Liu et al. developed aptamers for rhBMP-2 and experimented their applicability in ECM Mimetic Scaffolds. The manuscript includes prolific data of sequences, KD values, and secondary structures of rhBMP-2 aptamers, which are critical to the binding affinity and binding mechanisms between rhBMP-2 and its aptamers. I would recommend publishing this manuscript after some minor revisions.
1. What method was used to calculate binding affinity (KD values) in Figures 1, S8, and S10? This method and its formula KD = f(C) should be mentioned and explained in the manuscript.
2. The table summarizing sequences of selected aptamer candidates should be placed in the main manuscript. Additionally, their corresponding KD values should be added into this table.
3. What is the novelty and originality of the aptamers developed in this study compared to the published ones? Including a table comparing various aptamers for human bone morphogenetic protein 2 in terms of development method, sequence, KD, etc. can improve the quality of the manuscript.
4. Figure 1C: Why are the absorbance values at 450 nm negative?
Author Response
Reviewer 2:
"The research by Liu et al. developed aptamers for rhBMP-2 and experimented their applicability in ECM Mimetic Scaffolds. The manuscript includes prolific data of sequences, KD values, and secondary structures of rhBMP-2 aptamers, which are critical to the binding affinity and binding mechanisms between rhBMP-2 and its aptamers. I would recommend publishing this manuscript after some minor revisions.
1. What method was used to calculate binding affinity (KD values) in Figures 1, S8, and S10? This method and its formula KD = f(C) should be mentioned and explained in the manuscript.
Response: The formula used for KD value calculation was
Y =
X is the concentration of the ligand;
Y is the specific binding (binding percent);
Bmax is the maximum plateau value, expressed in the same units as the Y-axis;
KD is the equilibrium dissociation constant, expressed in the same units as the X-axis (concentration). When the drug concentration equals KD, half the binding sites are occupied at equilibrium.
KD value in this study was calculated by GraphPad Prism 8 using the non-linear fit model of One Site binding (hyperbola) that adopts the formula described above. The details were explained in Section 3.16 (Statistics and Data Analysis).
2. The table summarizing sequences of selected aptamer candidates should be placed in the main manuscript. Additionally, their corresponding KD values should be added into this table.
Response: No problem, we moved the table of sequences to the main text as Table 1. The KD values were also added to the table as suggested.
3. What is the novelty and originality of the aptamers developed in this study compared to the published ones? Including a table comparing various aptamers for human bone morphogenetic protein 2 in terms of development method, sequence, KD, etc. can improve the quality of the manuscript.
Response: Thank you, and we think this is an excellent suggestion. The novelty of this study is that this is the first reported specific aptamer for human bone morphogenic protein 2. Furthermore, the work may open exciting therapeutic avenues for bone damage by providing a potentially more efficient aptamer-functionalized scaffold for BMP-2 treatment. We so far haven’t found any nucleic acid aptamers for this target. Instead, an affinity peptide was reported earlier for BMP-2. The advantage of DNA aptamers over the affinity peptides was described in the Introduction section and we also highlighted that below,
“An earlier BMP-2-specific peptide (NH2-TSPHVPYGGGS-COOH) was reported in 2004 using a phage display [16]. Nucleic acid aptamers have antibody-like properties regarding specific molecular recognition of biomolecular surfaces, furthermore, may have particular advantages in terms of control of their function, biosafety, ease of assembly and low selection/manufacturing costs [17].”
4. Figure 1C: Why are the absorbance values at 450 nm negative?"
Response: Thanks for your question. The absorbance values at 450 nm were normalized by subtracting the background signals that were generated by the immobilized plate wells. The test sample with low binding to the aptamer may generate low absorbance that sometimes might be even lower than the background due to expected detection variations, which would lead to negative absorbance values after the normalization.
